# Cerebral and Splanchnic Vein Thrombosis: Advances, Challenges, and Unanswered Questions

**DOI:** 10.3390/jcm9030743

**Published:** 2020-03-10

**Authors:** Nicoletta Riva, Walter Ageno

**Affiliations:** 1Department of Pathology, Faculty of Medicine and Surgery, University of Malta, Msida MSD2080, Malta; nicoletta.riva@um.edu.mt; 2Department of Anatomy, Faculty of Medicine and Surgery, University of Malta, Msida MSD2080, Malta; 3Department of Medicine and Surgery, University of Insubria, 21100 Varese, Italy

**Keywords:** cerebral vein thrombosis, direct oral anticoagulants, splanchnic vein thrombosis, vitamin K antagonists

## Abstract

Cerebral vein thrombosis (CVT) and splanchnic vein thrombosis (SVT) are two manifestations of venous thromboembolism (VTE) at unusual sites. They have an incidence at least 25–50 times lower than usual site VTE, but represent true clinical challenges. Recent evidence on the epidemiology, risk factors, prognosis, and treatment of CVT and SVT has been published in the last two decades, thus contributing to a better understanding of these diseases. The improvement in imaging techniques and a higher degree of clinical suspicion may have led to the observed increased frequency, whereas a better knowledge of provoking mechanisms could have contributed to reducing the proportion of events classified as unprovoked or idiopathic (13–21% of CVT, 15–27% of SVT). Few small randomized clinical trials and a number of observational studies, although hampered by heterogeneous therapeutic approaches, shed light on the safety and effectiveness of anticoagulant therapy in these populations. However, there are still some grey areas that warrant future research. In this narrative review, we discuss recent advances and therapeutic challenges in CVT and SVT.

## 1. Introduction

Venous thromboembolism (VTE) can potentially occur in any venous segment. The most common disorders are represented by deep vein thrombosis (DVT) of the lower limbs and pulmonary embolism (PE). Unusual manifestations of VTE include DVT of the upper extremities, but also splanchnic, renal, ovarian, cerebral, and retinal veins thrombosis. Unusual site VTE has an incidence at least 25–50 times lower than usual site VTE, but often represent a true clinical challenge. Unfortunately, there is paucity of large clinical trials in the literature and, thus, solid evidence to drive patient management. In this narrative review, we will discuss recent advances and therapeutic challenges in cerebral vein thrombosis (CVT) and splanchnic vein thrombosis (SVT), starting from the presentation of two clinical cases.

## 2. Clinical Case No. 1

A 40-year-old woman presented to the Emergency Department because of a few days history of intense headache, associated with nausea, vomiting, drowsiness, and slurred speech. Past medical history was unremarkable. She had recently started a combined estrogen-progestin oral contraceptive pill. Cerebral magnetic resonance (MR) venography showed complete occlusion of the right transverse and sigmoid sinuses, with a small intraparenchymal hemorrhage. All blood test results were normal. She was started on weight-adjusted low molecular weight heparin (LWMH). A follow-up MR performed after a week showed improvement of the hemorrhage. LWMH was overlapped with a vitamin K antagonist (VKA), aiming for an international normalized ratio (INR) target range of 2.0–3.0.

## 3. Cerebral Vein Thrombosis

### 3.1. Definition and Epidemiology

CVT refers to thrombosis occurring in the cerebral veins and the dural venous sinuses [1]. The most common sites of thrombosis are the superior sagittal sinus (37–62%) and the lateral sinuses (31–44%) [2,3]. CVT is responsible for around 1% of all strokes [4].

Although the incidence of CVT was initially reported to be 3–4 cases per million adults [5], more recent studies reported 13.2–15.7 cases per million persons per year [6,7], and this increase is probably correlated with the progress in radiological investigations. CVT is more common in women, with a female/male ratio of about 3:1, and a mean age of 40 years [2,3,6].

The incidence of pediatric CVT is around 7 cases per million children, being particularly frequent in neonates and infants in the first year of age, without any specific sex predominance [8].

### 3.2. Risk Factors

CVT can be secondary to local or systemic risk factors, and in >40% of cases is a multifactorial disorder [1,2]. Local risk factors for CVT mainly include infections (8–12% of patients), such as infections of the ears, sinuses, mouth, face, neck, or the central nervous system (CNS) [2,3]. Thrombosis of the transverse and sigmoid sinuses are commonly described in patients with otitis or mastoiditis, whereas infections of the paranasal sinuses usually result in thrombosis of the cavernous sinuses [5]. Extremely rare forms of septic dural sinus thrombosis have also been described [9]. Mechanical causes are found in 2–5% of patients and include, amongst others, head trauma, lumbar puncture, jugular vein catheterization, and neurosurgical interventions [2,3]. Finally, CNS malignancies and other CNS disorders (e.g., arteriovenous malformation or dural fistulae) have been reported, each accounting for around 2% of patients [2].

Systemic risk factors are commonly found in CVT. Sex-specific risk factors, such as pregnancy/puerperium and hormonal treatments (such as oral contraceptives or estrogen replacement treatment), are reported in 10–17% and 50–53% of women with CVT, respectively [2,3]. Thrombophilia is another important risk factor for CVT, responsible for approximately a third of cases [2]. A recent meta-analysis reported a strong association, although derived from observational studies only, between the development of CVT and several thrombophilic abnormalities, such as factor V Leiden (odd ratio (OR) 2.89, 95% CI 2.10–3.97), prothrombin G20210A mutation (OR 6.05, 95% CI 4.12–8.90), antithrombin deficiency (OR 3.75, 95% CI 1.02–13.82), protein C deficiency (OR 8.35, 95% CI 2.61–26.67), protein S deficiency (OR 6.45, 95% CI 1.89–22.03), and hyperhomocysteinemia (OR 2.99, 95% CI 1.32–6.75) [10]. Data on antiphospholipid antibodies in CVT patients are scarce. The combined presence of thrombophilic abnormalities and oral contraceptive use can further increase the risk of developing CVT [11]. Finally, although the presence of the V617F mutation in the Janus kinase 2 gene (JAK2V617F) has been identified in 6.6% of CVT patients [12], only 3.8% had a diagnosis of myeloproliferative neoplasm (MPN) throughout their lives [13]. It has been suggested not to routinely screen all CVT patients for occult malignancy, due to the low prevalence of this condition, or thrombophilia, given the limited clinical relevance. However, in accurately selected patients (such as those with young age, unprovoked CVT, or family history of VTE) [14] finding a severe thrombophilia (i.e., antithrombin, protein C or S deficiencies, or homozygosity for factor V Leiden or prothrombin G20210A mutation) would suggest an indefinite anticoagulant treatment duration, similarly to usual site VTE.

The risk factors for pediatric CVT were different between neonates and non-neonates—acute systemic illnesses were more frequent in neonates (84% vs. 31%, respectively), whereas systemic diseases (4% vs. 60%), prothrombotic states (20% vs. 54%), and head/neck disorders (16% vs. 38%) were more common in infants and children [8]. Lastly, in approximately 13–21% of CVT, no risk factor can be identified and these cases are classified as unprovoked [2,15].

### 3.3. Clinical Presentation

Symptoms of CVT are often not specific, and a median delay of 7 days (mean ± SD, 18.3 ± 59.4 days) was reported between the onset of symptoms and CVT diagnosis [2].

The first and most commonly reported symptom is headache (87–88% of patients) [2,16,17]. The headache is usually described as a diffuse and progressively increasing pain, typically exacerbated by the recumbent position and the Valsalva maneuver [17]. Headache can be the only symptom in up to a quarter of patients [16]. Approximately 10% of patients do not manifest headache at CVT onset, and this presentation is more common in males and in older people [18]. Headache is frequently associated with other signs of intracranial hypertension, such as papilloedema, which can lead to permanent visual loss if not promptly treated [5]. In 19–23% of patients, CVT can present with seizures, which can be focal, focal with secondary generalization, or generalized tonic–clonic seizures, and may evolve into status epilepticus [2,16,19].

CVT can potentially mimic an ischemic stroke, however, headache and seizures are more common than in arterial events [1]. Focal neurological deficits are usually secondary to the presence of a venous infarction, and are more common in CVT of the superficial cerebral veins [1]. Aphasia can occur in thrombosis of the lateral sinus of the dominant hemisphere. Coma and bilateral neurological focal deficits have been reported in patients with occlusion of the deep cerebral veins [4].

### 3.4. Diagnosis

A recent study reported that in approximately 1 every 30 patients there was a CVT misdiagnosis, defined as attendance at the emergency department for headache or seizure within 2 weeks prior to the actual CVT diagnosis [20].

There is no clinical prediction rule currently available and D-dimer still plays a limited role. A meta-analysis of 14 studies reported a weighted mean sensitivity of 93.9% and specificity of 89.7% for D-dimer [21]. However, false negative D-dimer levels were common in patients with isolated headache or prolonged duration of symptoms (e.g., >1 week) [21]. Although with a weak strength of recommendation, current guidelines suggest D-dimer testing before imaging investigations, except in these two categories of patients [14].

In patients with clinical suspicion of CVT, urgent neuroimaging should be performed. Plain (unenhanced) computed tomography (CT) can show direct signs of CVT (such as the “dense triangle” or the “cord sign”); however, these signs are present only in approximately 30% of patients and their absence is not sufficient to rule out CVT [1]. CT venography can confirm CVT diagnosis by showing the intraluminal filling defects, and has high sensitivity for both dural venous sinus thrombosis (99%) and cerebral vein thrombosis (88%) [1]. CT venography is usually widely available; however, some anatomic variants (such as sinus hypoplasia or atresia) can be misclassified as sinus thrombosis, whereas cortical vein thrombosis can remain undetected by CT venography alone [22]. MR is nowadays the reference standard imaging for CVT, especially when performed as contrast-enhanced MR venography [22]. However, cerebral MR requires good patient cooperation and long time to complete the scan. When MR venography is not available, CT venography can be used as a reliable alternative [14]. Intra-arterial angiography (or digital subtraction angiography) was once the gold standard for CVT diagnosis, but it is currently executed only if non-invasive imaging is inconclusive or to perform endovascular procedures [22].

### 3.5. Prognosis

CVT has generally a good outcome, with low rates of mortality and residual disability. The improvement in diagnostic techniques, with the possible diagnosis of less severe CVT episodes, increased awareness of disease, and improved therapeutic strategies has resulted in a reduced mortality rate over time [23]. Recent studies reported a mortality rate at discharge between 1% and 9% and a mortality rate during follow-up between 3% and 12% [2,6,7]. Mortality during the acute phase was mainly due to direct CVT complications, such as transtentorial herniation, whereas underlying conditions contributed to mortality rate during follow-up [24]. Onset with seizures did not appear to be a negative prognostic factor [19].

The presence of intracranial hemorrhage at the time of CVT diagnosis was not an infrequent finding, reported in 20–39% of patients [2,3,16,25] and was a negative prognostic factor [16,25,26].

More than 80% of CVT patients achieved recanalization, either complete or partial, which was associated with a good functional recovery [27,28]. However, it is still unclear whether the lack of recanalization should influence anticoagulant treatment duration, as the evidence on its correlation with CVT recurrence was scarce [28].

Only 5–10% of CVT survivors showed residual disability or dependence [1]. A prognostic score (the CVT risk score) has been proposed to identify patients with a poor prognosis [29]. Six variables were included (malignancy, coma, deep cerebral vein thrombosis—2 points each, altered mental status, male sex, intracranial hemorrhage—1 point each) and a score ≥ 3 points showed a sensitivity of 96.1% for death or dependency [29].

Approximately 3% of CVT patients had a recurrent CVT, whereas an additional 7% had a DVT and/or PE, corresponding to an incidence rate of recurrent VTE of 2.03 per 100 person-years [15]. Most of recurrent thrombotic events occurred in the first year after anticoagulant treatment discontinuation [15].

### 3.6. Treatment

The latest guidelines on the treatment of adult patients with CVT were released by the European Stroke Organization (ESO), endorsed by the European Academy of Neurology (EAN), and published in 2017 [14]. They recommended anticoagulant treatment with therapeutic heparin during the acute phase of CVT, despite the presence of intracerebral hemorrhage (strong recommendation) [14]. They suggested starting with low molecular weight heparin (LMWH) instead of unfractionated heparin (UFH), if no contraindication or planned surgery, and to continue with VKA (weak recommendations). The direct oral anticoagulants (DOAC) were not recommended for CVT treatment (weak recommendation) at the time of publication of these guidelines [14], however, further evidence has been published in the last few years.

The treatment of pediatric CVT was mentioned in the guidelines of the American Society of Hematology published in 2018 [30], which recommended anticoagulation in CVT without concomitant hemorrhage (strong recommendation) and suggested anticoagulation in CVT with hemorrhage (conditional recommendation).

Heparin, either UFH or LMWH, is largely prescribed for the acute phase of CVT. However, evidence on its efficacy and safety was mainly derived from a meta-analysis of two small randomized controlled trials (RCT), which reported that both UFH and LWMH, when compared to placebo, were associated with a trend towards a reduction of death (relative risk (RR) 0.33, 95% CI 0.08–1.21) [31]. More recently, LMWH was compared to UFH in another two small RCTs and showed a tendency towards lower rates of mortality (odd ratio (OR) 0.21, 95% CI 0.02–2.44) or severe disability (OR 0.50, 95% CI 0.11–2.23) [32].

Thrombolysis should not be considered a routine treatment of CVT [14], due to the high risk for major bleeding complications (weighted mean rate 9.8%), which were mainly intracranial hemorrhages (weighted mean rate 7.6%) and carried a high fatality rate (58.3% of intracranial hemorrhages) [33]. However, thrombolysis may be considered as a second line option in CVT causing progressive clinical deterioration despite anticoagulant treatment [34,35].

After the initial parenteral treatment, most of the patients are switched to VKA. Three large observational studies provided outcome data on CVT patients treated with the conventional anticoagulant treatment (UFH or LMWH, eventually followed by VKA). The International Study on Cerebral Vein and Dural Sinus Thrombosis (ISCVT) included 624 CVT patients, of whom 83% were anticoagulated during the acute phase [2]. During a median follow-up of 16 months, this subgroup of patients showed a trend towards lower rates of death or dependency (12.7% vs. 18.3%, hazard ratio (HR) 0.73, 95% CI 0.44–1.21) [2]. The Cerebral Vein Thrombosis International Study (CEVETIS) included 706 CVT patients, of whom 85% were treated with heparin in the acute phase and 84% were prescribed with VKA for a median duration of 12 months [3]. During a median follow-up of 40 months, CVT prognosis in general was good, with 89.1% of patients having a complete recovery and 3.8% being independent despite a partial recovery [3]. More recently, a multicenter Turkish study (VENOST) included 1144 CVT patients, of whom 84% received heparin during the acute phase and 67% continued with warfarin [16]. One-year follow-up data were available for 691 patients, and 93.1% of them had a complete recovery [16].

The main studies evaluating the use of the DOACs in patients with CVT are summarized in Table 1. They were mainly small observational studies, either retrospective [36,37,38,39,40] or prospective [41,42,43], and there was only one RCT [44]. The number of patients treated with DOAC ranged from 6 [38] to 60 [44] patients. The average anticoagulant treatment duration ranged from 5 [44] to 12 [40] months. Three studies evaluated rivaroxaban [36,38,41], two dabigatran [37,44], and three cohorts included different DOACs [40,42,43]. In the majority of the studies, the DOAC were used after an initial treatment with UFH or LMWH [36,37,40,42,44] in order to achieve stable conditions. There was only one study enrolling 20 patients who started directly with rivaroxaban 15 mg twice daily (BID) for 3 weeks, followed by 20 mg once daily (OD) [41]. The RE-SPECT CVT trial (A Clinical Trial Comparing Efficacy and Safety of Dabigatran Etexilate With Warfarin in Patients With Cerebral Venous and Dural Sinus Thrombosis) randomized 120 CVT patients to either dabigatran 150 mg BID or dose-adjusted warfarin (INR target range 2.0–3.0) after an initial parenteral treatment with either UFH or LMWH for 5–15 days [44]. However, patients with central nervous system infections, major head trauma, active cancer, or coma were excluded from this trial [44].

Taken together, the results of these studies showed that the rates of recurrent CVT and VTE were low during anticoagulant treatment. The use of the DOACs was associated with extremely variable rates of excellent neurological outcomes (64.1–100% of patients) and major bleeding events (0–8.3% of patients); thus, larger studies are needed to support these findings.

The 2017 ESO guidelines suggested a variable anticoagulant treatment duration between 3 and 12 months (weak recommendation), which can be prolonged in patients with recurrent VTE or persistent prothrombotic conditions [14]. Previous guidelines, released by the European Federation of the Neurological Societies (EFNS) in 2010 [35] and the American Heart Association/American Stroke Association (AHA/ASA) in 2011 [34], suggested 3-6 months of anticoagulant treatment for CVT secondary to transient risk factors, 6-12 months for unprovoked CVT, and indefinite duration for recurrent CVT or VTE or severe thrombophilia. These recommendations are supported by the non-negligible risk of recurrent VTE [15] and are partly derived from the treatment of usual site VTE [45]. There is an ongoing randomized study, the EXCOA-CVT study (The benefit of EXtending oral antiCOAgulation treatment after acute cerebral vein thrombosis), comparing short (3–6 months) vs. long term (12 months) anticoagulant treatment duration for CVT [46].

## 4. Clinical Case No. 2

A 70-year-old man presented to the emergency department because of severe abdominal pain and vomiting. Abdomen X-ray was negative for free air or fluids levels. Abdomen CT venography showed thrombosis of the superior mesenteric vein extending to the confluence of the portal vein, as well as edema of the small bowel walls. Anticoagulation was started with UFH, and after a few days of clinical stability the patient was switched to LMWH. Warfarin was introduced a week later, aiming to INR target range 2.0–3.0. A 1-month follow-up abdominal CT scan showed complete recanalization of the superior mesenteric vein. Anticoagulation was continued lifelong.

## 5. Splanchnic Vein Thrombosis

### 5.1. Definition and Epidemiology

SVT refers to thrombosis occurring in the veins draining the abdominal viscera, and includes portal vein thrombosis (PVT), mesenteric veins thrombosis (MVT), splenic vein thrombosis, and Budd–Chiari syndrome (BCS) [47]. BCS refers to the obstruction of the hepatic venous outflow, which can be located at any site between the small hepatic venules and the confluence of the inferior vena cava into the right atrium [48].

The epidemiology of SVT differs according to involved veins. The highest incidence rates were reported for PVT (1.73 and 3.78 per 100,000 persons per year in females and males, respectively [49]), whereas BCS showed very low incidence rates (around 1–2 cases per million persons per year [49,50]). MVT was more frequently reported in older people in their sixties/seventies [51,52], whereas patients with BCS were usually in their thirties/forties [53,54]. Finally, PVT appeared to be more common in males (male/female ratio around 1.5–2:1) [49,54], whereas a slightly predominance of females has been reported for BCS (male/female ratio around 1:1.3-2) [53,55].

### 5.2. Risk Factors

It was initially reported that SVT were more commonly secondary to systemic risk factors [56,57]; however, recent data highlighted that liver cirrhosis and solid abdominal cancer account for approximately half of the cases [54,58] and that SVT is a multifactorial disorder in 46–64% of patients [59,60].

Among local risk factors for SVT, liver cirrhosis is responsible for 24–27% of SVT [54,58], whereas the prevalence of SVT in cirrhosis was reported to be 17%, although asymptomatic and incidentally detected in 43% of them [61]. Solid cancer is found in 22–27% of SVT [54,58] and, in addition, approximately 8% of SVT patients are diagnosed with cancer within the first 3 months of follow-up [62]. Other local risk factors are recent abdominal surgery and abdominal infections or inflammation (e.g., pancreatitis, cholecystitis, appendicitis, diverticulitis, liver abscesses, inflammatory bowel diseases). Rare forms of septic thrombosis of the portal venous system (pylephlebitis) have also been described [63].

MPN represent the most common systemic risk factor, being present in 40.9% of BCS and 31.5% of non-malignant non-cirrhotic PVT [64,65]. In particular, the JAK2V617F mutation was reported in 37–45% of BCS and 24–34% of PVT [66,67]. Paroxysmal nocturnal hemoglobinuria (PNH) is another hematological disorder that can predispose to the development of SVT, particularly BCS [68]. However, the prevalence of PNH among SVT patients is <1% [69].

The role of thrombophilic disorders has been recently evaluated by several systematic reviews. A strong association was reported between the presence of factor V Leiden and the development of BCS (OR 6.21, 95% CI 3.93–9.79), non-cirrhotic PVT (OR 1.85, 95% CI 1.09–3.13), and cirrhotic PVT (OR 2.55, 95% CI 1.29–5.07), whereas prothrombin G20210A mutation was associated only with non-cirrhotic PVT (OR 5.01, 95% CI 3.03–8.30) [70]. A strong association was also reported for the development of PVT and the presence of antithrombin deficiency (OR 8.89, 95% CI 2.34–33.72), protein C deficiency (OR 17.63, 95% CI 1.97–158.21), and protein S deficiency (OR 8.00, 95% CI 1.61-39.86) [71]. There were only a few data on natural anticoagulant deficiencies in BCS compared to healthy controls [71], whereas the role of antiphospholipid antibodies still needs to be better elucidated [72]. Among systemic risk factors, hormonal stimuli also have been reported (such as oral contraceptives, hormone replacement therapy, pregnancy/puerperium), which play an important role, especially for BCS [60].

Thorough investigation of all SVT patients for the presence of major local (such as solid abdominal cancer or liver cirrhosis) and systemic risk factors (such as MPN) has been suggested, given the prognostic and treatment implications of these findings [73,74]. In particular, screening for the JAK2V617F mutation should be performed in all patients without major local risk factors [75]. After excluding major local and systemic risk factors, thrombophilia testing is suggested in patients with high pretest probability of severe thrombophilia (such as those with young age, unprovoked SVT, or family history of VTE).

However, in around 15–27% of SVT, no risk factor can be identified, and these cases are classified as unprovoked SVT [54,58].

### 5.3. Clinical Presentation

Symptoms of SVT are often not specific, and a median delay of 7 days (interquartile range 3–18) was reported between the onset of symptoms and SVT diagnosis [76]. Most of the patients present with abdominal pain (48–55%), ascites (10–29%), and/or gastrointestinal bleeding (9–26%) [54,58]. However, 18–30% of patients are asymptomatic, and SVT can be an incidental finding at abdominal imaging performed for other clinical indications [54,58,77,78,79].

In particular situations, there are also more specific manifestations of SVT. For instance, chronic PVT shows portal cavernoma formation and symptoms of portal hypertension, such as splenomegaly, thrombocytopenia, gastroesophageal varices, portal cholangiopathy, and hepatic encephalopathy [60]. Acute MVT is characterized by a severe abdominal pain, which often seems to be excessive compared to physical examination findings. Non-specific symptoms, such as nausea, vomiting, and gastrointestinal bleeding are also frequently reported [80]. A typical triad has been described for patients with BCS, consisting of hepatomegaly, ascites, and abdominal pain. However, there are also rare forms of fulminant BCS with early onset of hepatocellular necrosis and rapid liver failure [60].

### 5.4. Diagnosis

There is no clinical prediction rule currently available for SVT and the utility of D-dimer in this setting is still controversial. Although there are some small studies showing higher D-dimer levels in SVT patients [81,82], others reported a correlation with the severity of liver disfunction [83,84].

Doppler ultrasound (US) is the first line imaging for the diagnosis of PVT (sensitivity 89–93%, specificity 92–99%) [73,85]. Abdominal CT scan or MR are suggested to confirm the extent of PVT [73], as the sensitivity of US for MVT is suboptimal (70–90%), mainly due to difficulty visualizing the mesenteric veins [85]. Therefore, CT venography and MR are the imaging of choice for the diagnosis of MVT (sensitivity 91–95% and specificity 94–100% for CT, sensitivity and specificity approximately 100% for MR) [74,85]. CT scan is more readily available than MR, and can also visualize the bowel walls and show signs of small bowel ischemia [86]. Digital subtraction angiography is nowadays performed only when endovascular procedures are needed or when non-invasive imaging tests are inconclusive.

Doppler US is also the first line imaging for BCS, particularly when performed by experienced technicians (specificity and sensitivity approximately 85–90%) [73,87], even though CT or MR are required for confirmation [73]. Liver biopsy is not required for BCS diagnosis, but can be useful to confirm the diagnosis when only the small intrahepatic veins are involved or to rule out other hepatic disorders [87,88].

### 5.5. Prognosis

The mortality rate reported after an episode of SVT is higher than in the general population, both in the short- (30 days) and long-term (up to 20 years). In a Danish cohort study, the 30-day risk of mortality for SVT patients was 20.6% vs. 0.7% for the comparison cohort, matched for a number of comorbidities, resulting in a mortality rate ratio of 40.7 (95% CI 32.4–51.1) [52]. In particular, MVT was associated with the highest short-term mortality rate (63.1% at 30 days), whereas PVT showed the highest mortality rate in the following period (e.g., 23.1% at 1 year, 27.2% at 5 years) [52]. This finding can be explained by the fact that MVT can be complicated by acute bowel infarction in around a third of patients [51].

The incidence of arterial or venous thrombotic events after SVT in a multicentre international unselected cohort was 7.3 per 100 patient-years (95% CI 5.8–9.3) [58]. Some differences emerged on the basis of the pathogenesis of SVT, with the risk of further thrombotic events being the highest in patients with liver cirrhosis (11.3 per 100 patient-years), followed by solid cancer (7.6 per 100 patient-years), unprovoked SVT (6.3 per 100 patient-years), MPN (5.9 per 100 patient-years), and SVT associated with transient risk factors (3.2 per 100 patient-years) [58]. A recent cohort study of 181 patients with MPN and SVT highlighted that the rate of thrombotic complications is quite high even during VKA treatment (3.9 per 100 patient-years) [89].

Data from Danish registries showed that SVT patients had higher risk of bleeding compared to patients with usual site VTE and with the general population, both in the short- (30 days) and long-term (up to 10 years). The 30-day risk of bleeding for SVT patients was 4.3% vs. 0.5% for usual site VTE (adjusted HR 9.64, 95% CI 6.46–14.40) and 0.1% for the general population (adjusted HR 39.79, 95% CI 19.44–81.46) [90]. A higher incidence of arterial cardiovascular events was also reported, and the 30-day risk was 3.3% for SVT patients vs. 0.9% for usual site VTE (adjusted HR 7.05, 95% CI 4.74–10.48) and 0.4% for the general population (adjusted HR 15.75, 95% CI 9.26–26.79) [90].

### 5.6. Treatment

The latest guidelines on the treatment of patients with SVT were published by the European Association for the Study of the Liver (EASL) in 2016 [73]. In patients with non-malignant non-cirrhotic PVT, they recommended starting anticoagulation immediately with LMWH, if no major contraindication, and to continue with VKA (INR target range 2.0–3.0) for at least 6 months (strong recommendations) [73]. In cirrhotic patients, adequate prophylaxis of gastrointestinal bleeding should be instituted before starting anticoagulation (strong recommendation) [73]. Initial treatment of patients with BCS consists of anticoagulation; however, in non-responsive patients with progressive liver deterioration, a stepwise approach should be followed: (1) medical treatment, (2) endovascular procedures (e.g., angioplasty, stenting, thrombolysis), (3) transjugular intrahepatic portosystemic shunt (TIPS), and (4) liver transplant [73].

The treatment of MVT was also mentioned by the guidelines of the European Society of Vascular Surgery (ESVS), published in 2017 [74]. Anticoagulation should be the first line treatment of MVT, starting with LMWH or preferably UFH, because in the early phase a laparotomy for bowel necrosis might be required. Antibiotics are indicated if there is a septic thrombophlebitis or sepsis due to bowel perforation or bacterial translocation. About 5% of patients deteriorate despite medical treatment and they are candidates for endovascular procedures (such as thrombolysis, angioplasty, thrombectomy, or TIPS) [74]. Switching to oral anticoagulants, either VKA or DOAC, is suggested after the acute phase (2–3 weeks), unless there is a strong preference for parenteral anticoagulation (e.g., paraneoplastic thrombosis) [74].

Although initial studies highlighted the high incidence rates of gastrointestinal bleeding (12.5 per 100 patient-years) of SVT patients [91] and an approximately twofold increased bleeding risk associated with anticoagulant treatment [54,92], more recent data from large multicenter cohort studies contributed to a better understanding of the safety and effectiveness of anticoagulant treatment in this population. The prospective International Registry on Splanchnic Vein Thrombosis (IRSVT) enrolled 604 SVT patients, of whom 465 (77.0%) received anticoagulant treatment with either parenteral anticoagulation only (175 patients, 37.6%) or VKA (290 patients, 62.4%) [58]. The incidence of major bleeding events was low (3.8 per 100 patient-years overall, 3.9 per 100 patient-years on treatment, 5.8 per 100 patient-years in untreated patients), and the time on anticoagulant treatment was associated with a lower risk of bleeding events (HR 0.90, 95% CI 0.84–0.96) [58]. In another multicenter cohort of 375 SVT patients treated with VKA, the risk of major bleeding event was 1.24 per 100 patient-years [93].

Cirrhotic patients represent a particular category of patients because they are associated with the highest risk of major bleeding and recurrent thrombotic events [58,94]. However, this increased risk of bleeding should not prevent anticoagulant treatment for these patients, as a meta-analysis of eight studies reported no significant difference in the rate of any bleeding between treated and untreated cirrhotic patients with PVT (11% in both groups) [95] and a recent study on 182 cirrhotic patients with PVT showed similar rates of major bleeding events among treated and untreated patients (21.8% and 19.7%, respectively) [96].

The main studies evaluating the use of the DOACs in patients with SVT are summarized in Table 2. These were mainly small observational studies, either retrospective [97,98,99,100,101,102] or prospective [103], with only one RCT [104]. The number of patients treated with DOAC ranged from 12 [98] to 93 [102] patients. The average anticoagulant treatment duration ranged from 6 [99] to 10.8 [103] months. One study evaluated dabigatran [100], one edoxaban [99], and one rivaroxaban [104], and five cohorts evaluated different DOACs [97,98,101,102,103]. Three cohorts included patients treated with the DOACs for different clinical indications [97,98,103], and no separate data were reported for SVT patients. The only RCT was an open-label study that included 40 patients in the rivaroxaban arm (10 mg BID) and 40 patients in the warfarin arm (INR target range 2.0–2.5), and both groups initially received 3 days of therapeutic LMWH [104]. Rivaroxaban was associated with higher rates of vessel recanalization and lower rates of SVT recurrence or major bleeding; however, these data should be interpreted with caution. Taken together, the results of these studies suggest extremely variable rates of recurrent SVT (0–11.1%), major bleeding (0–9.1%), and recanalization (68.8–100%), which can be partly explained by the large heterogeneity among the different cohorts. There is an ongoing interventional study evaluating the use of rivaroxaban in patients with non-cirrhotic SVT (NCT02627053).

The 2016 EASL guidelines suggested considering an anticoagulant treatment duration of at least 6 months to be extended lifelong in patients with BCS, strong prothrombotic conditions (including persistent risk factors, such as MPN), recurrent VTE, intestinal ischemia or involvement of the superior mesenteric vein, and in cirrhotic patients eligible for liver transplant [73]. Similar recommendations were provided by the Baveno VI consensus published in 2015 [88], whereas the 2017 ESVS guidelines suggested considering also the potential consequences of a recurrent SVT (e.g., short bowel syndrome) when deciding about anticoagulant discontinuation [74]. More general guidelines on the antithrombotic treatment of VTE recommend a minimum anticoagulant treatment duration of 3 months for a thrombosis of the portal venous system secondary to transient risk factors, while suggesting extended anticoagulation for unprovoked SVT or SVT secondary to permanent risk factors [45].

## 6. Expert Opinion

In general, we start anticoagulant treatment for CVT in the acute phase with LMWH (or UFH in patients with severe renal failure). UFH may also be preferred in patients with intracranial hemorrhage and severe clinical presentation. After clinical stability, the majority of patients are switched to VKA. Pending more evidence on the use of the DOACs and on the basis of the favorable results of the RE-SPECT CVT trial [44], we consider the use of dabigatran in selected patients with good general health conditions and low bleeding risk. However, the DOACs are currently off-label for the treatment of unusual VTE. A minimum anticoagulant treatment duration of 3 months is usually prescribed, which can be extended up to 6–12 months for unprovoked CVT and become indefinite for recurrent VTE or persistent risk factors (such as severe thrombophilia).

After assessing the risk of bleeding, we usually start the anticoagulant treatment for SVT in the acute phase with LMWH, together with adequate prophylaxis of variceal bleeding, if needed. In patients with thrombocytopenia, a reduced dose of LMWH might be required. We consider UFH in patients with recent bleeding, severe renal failure, or potential candidate for surgery (such as those with extensive bowel infarct). After clinical stability, we switch the majority of patients to VKA, with the exception of those with cancer-associated SVT, severe thrombocytopenia, or severe liver disease. Pending more evidence on the use of the DOACs and on the basis of the results of RCTs conducted on usual site VTE [105,106,107,108], we consider the use of factor-Xa inhibitors in selected patients without severe liver or renal impairment and at low bleeding risk. The majority of SVT patients are candidate for long-term anticoagulation, having either a major persistent risk factor, an unprovoked SVT, or a BCS; however, we favor a shorter duration of 3–6 months in those with transient risk factors (such as recent abdominal surgery or abdominal inflammation/infections).

## 7. Conclusions and Future Directions

Increasing evidence on the epidemiology, risk factors, prognosis, and treatment of CVT and SVT has been published in the last two decades, thus contributing to a better understanding of these disorders. It is well-known nowadays that CVT and SVT are not as “uncommon” as previously reported, even though their incidence remains at least 25–50 times lower than usual site VTE. The improvement in imaging techniques and a higher degree of clinical suspicion could have contributed to their increased frequency. The better knowledge of local and systemic risk factors, as well as the systematic search for them in some patients, has reduced the percentage of events classified as idiopathic or unprovoked to 13–21% of CVT and 15–27% of SVT. Few small randomized controlled trials and a number of observational studies, although hampered by heterogeneous treatment approaches, confirmed the relative safety and effectiveness of the conventional anticoagulant therapy in these populations.

Nevertheless, there are still some grey areas that need to be better explored by future research. For instance, no clinical algorithm has been proposed for CVT and SVT and the role of D-dimer still needs to be clarified. Although there seems to be some evidence to support its use in patients with suspected CVT, despite some false negative results, it is still not clear whether the increased values of D-dimer levels reported in SVT patients correlate mainly with the thrombotic event, or the degree of liver dysfunction, or both.

In addition, there is still a paucity of data on the safety and effectiveness of the DOACs in these populations, which were excluded from the large phase III RCTs conducted in patients with DVT and/or PE. The use of the DOACs in unusual site VTE is still off-label, but an increasing number of case reports and small case series have already been published. Given the rarity of unusual site VTE, collaborative studies are highly warranted. There is an ongoing prospective international registry (NCT03778502) that will provide real-life evidence on the use of the DOACs in patients with CVT, SVT, and other unusual site VTE.

## Figures and Tables

**Table 1 jcm-09-00743-t001:** Studies evaluating the use of the direct oral anticoagulants in patients with cerebral vein thrombosis.

Author (year)	Study Design	No. Patients	Demographics	Treatment	Treatment Duration	Safety and Efficacy Outcomes
**Geisbüsch (2014)** **[36]**	Retrospective	7	Age (median): 31 years Female/male 7:0	Rivaroxaban (15 mg BID, followed by 20 mg OD, or directly 20 mg OD), both preceded by heparin	8 months (median)	Excellent outcome (mRS 0–1): *n* = 7 (100%) Recanalization (partial or complete): *n* = 7 (100%) Recurrent VTE: *n* = 0 (0%) Major bleeding: *n* = 0 (0%) Minor bleeding: *n* = 2 (28.6%)
		9	Age (median): 43 years Female/male 6:3	Phenprocoumon (INR target range 2.0–3.0), preceded by heparin	9 months (median)	Excellent outcome (mRS 0–1): *n* = 8 (88.9%) Recanalization (partial or complete): *n* = 9 (100%) Recurrent VTE: *n* = 0 (0%) Major bleeding: *n* = 0 (0%) Minor bleeding: *n* = 1 (11.1%)
**Mendonça (2015)** **[37]**	Retrospective	15	Age (median): 38 years Female/male 12:3	Dabigatran (110 or 150 mg BID), preceded by heparin/VKA	6 months (median)	Excellent outcome (mRS 0–1): *n* = 13 (86.7%) Recanalization: *n* = 12 (80%) Recurrent VTE: *n* = 0 (0%) Major or minor bleeding: *n* = 0 (0%)
**Anticoli (2016)** **[38]**	Retrospective	6	Age (mean): 36.5 years Female/male 6:0	Rivaroxaban (15 mg BID, followed by 20 mg OD, or directly 20 mg OD), the latter preceded by heparin/VKA	11 months (median)	Excellent outcome (mRS 0–1): *n* = 6 (100%) Recanalization (partial or complete): *n* = 6 (100%) Recurrent VTE: *n* = 0 (0%) Major or minor bleeding: *n* = 0 (0%)
**Herweh (2017)** **[39]**	Retrospective	13	Age (median): 38 years * Female/male 81:18 *	DOAC (unspecified), preceded by heparin	7 months (median) *	Recanalization (partial or complete): *n* = 11 (84.6%) Recurrent CVT: *n* = 0 (0%) Major bleeding: *n* = 0 (0%)
		86	Age (median): 38 years * Female/male 81:18 *	Phenprocoumon (INR target range 2.0-3.0), preceded by heparin (*n* = 83), or LMWH only (*n* = 3)	7 months (median) *	Recanalization (partial or complete): *n* = 75 (87.2%) Recurrent CVT: *n* = 0 (0%) Major bleeding: *n* = 0 (0%)
**Shankar Iyer(2018)** **[41]**	Prospective	20	Age (mean): 34.1 years Female/male 4:16	Rivaroxaban (15 mg BID, followed by 20 mg OD), without heparin	6 months (mean)	Excellent outcome (mRS 0–1): *n* = 19 (95%) Recanalization (partial or complete): *n* = 20 (100%) Recurrent VTE: *n* = 0 (0%) Major or minor bleeding: *n* = 0 (0%)
**Covut (2019)** **[40]**	Retrospective	9	Age (median): 56 years Female/male 7:2	Apixaban or rivaroxaban, preceded by heparin/VKA	12 months (median)	Recanalization (partial or complete): *n* = 5 (55.6%) Recurrent VTE: *n* = 0 (0%) Major or minor bleeding: *n* = 0 (0%)
**Ferro (2019)** **RE-SPECT CVT** **[44]**	Randomized controlled trial	60	Age (mean): 45.2 years Female/male 33:27	Dabigatran (150 mg BID), preceded by heparin	5.1 months (mean)	Excellent outcome (mRS 0–1): *n/N* = 54/59 (91.5%) Recanalization (partial or complete): *n/N* = 33/55 (60%) Recurrent VTE: *n* = 0 (0%) Major bleeding: *n* = 1 (1.7%) Clinically-relevant non-major bleeding: *n* = 0 (0%) Any bleeding: *n* = 12 (20%)
		60	Age (mean): 45.2 years Female/male 33:27	Warfarin (INR target range 2.0–3.0), preceded by heparin	5.3 months (mean)	Excellent outcome (mRS 0–1): *n/N* = 53/58 (91.4%) Recanalization (partial or complete): *n/N* = 35/52 (67.3%) Recurrent VTE: *n* = 0 (0%) Major bleeding: *n* = 2 (3.3%) Clinically-relevant-non-major bleeding: *n* = 1 (1.7%) Any bleeding: *n* = 12 (20%)
**Rusin (2019)** **[42]**	Prospective	36	Age (mean): 40.3 years Female/male 21:15	Apixaban (5 mg BID), dabigatran (110 or 150 mg BID), rivaroxaban (20 mg OD), all preceded by heparin	8.5 months (median)	Excellent outcome (mRS 0–1): *n* = 24 (66.7%) Recanalization (partial or complete): *n* = 34 (94.4%) Recurrent CVT: *n* = 2 (5.6%)—off anticoagulation Deep vein thrombosis: *n* = 2 (5.6%)—off anticoagulation Major bleeding: *n* = 3 (8.3%)
**Wasay (2019)** **[43]**	Prospective	45	Age (mean): 36.5 years Female/male 27:18	Dabigatran (75–150 mg BID), rivaroxaban (15–20 mg OD), 80% preceded by heparin	8 months (median)	Excellent outcome (mRS 0–1): *n/N* = 25/39 (64.1%) Recurrent VTE: *n* = 0 (0%) Major bleeding: *n* = 1 (2.2%) Any bleeding: *n* = 2 (4.4%)
		66	Age (mean): 41.3 years Female/male 37:29	Warfarin (INR target range 2.0–3.0), 65% preceded by heparin	8 months (median)	Excellent outcome (mRS 0–1): *n/N* = 35/56 (62.5%) Recurrent VTE: *n* = 0 (0%) Major bleeding: *n* = 1 (1.5%) Any bleeding: *n* = 4 (6.1%)

Legend: BID = twice daily; CVT = cerebral vein thrombosis; DOAC = direct oral anticoagulant; INR = international normalized ratio; mRS = modified Rankin scale; OD = once daily; VKA = vitamin K antagonist; VTE = venous thromboembolism; y = years. * data not separately reported for CVT patients treated with DOAC or other anticoagulants.

**Table 2 jcm-09-00743-t002:** Studies evaluating the use of the direct oral snticoagulants in patients with splanchnic vein thrombosis.

Author (year)	Study Design	No. Patients	Demographics	Treatment	Treatment Duration	Safety and Efficacy Outcomes
**Intagliata (2016)** **[98]**	Retrospective	20 (but only 12 SVT)	Age (median): 57 years * Female/male 10:10 * All liver cirrhosis	Apixaban (2.5–5 mg BID), rivaroxaban (10–20 mg OD)	8.8 months (median) *	Major bleeding: *n* = 1 (5%) * Any bleeding: *n* = 4 (20%) *
		19 (but only 6 SVT)	Age (median): 60 years * Female/male 7:12 * All liver cirrhosis	LMWH and/or VKA (INR target range not reported)	15.7 months (median) *	Major bleeding: *n* = 2 (10.5%) * Any bleeding: *n* = 3 (15.8%) *
**De Gottardi (2017)** **[97]**	Retrospective	94 (but only 69 SVT)	Age (mean): 55.4 years * Female/male 44:50 * 62% liver cirrhosis *	Dabigatran (110–220 mg daily), apixaban (2.5–10 mg daily), rivaroxaban (5–20 mg daily)	10.5 months (median) in patients without cirrhosis * 7.0 months (median) in patients with cirrhosis *	Recurrent SVT: *n* = 1 (1.1%) * Major bleeding: *n* = 3 (3.2%) * Minor bleeding: *n* = 11 (11.7%) *
**Janczak (2018)** **[103]**	Prospective	36 (but only 25 SVT)	Age (mean): 53.6 years * Female/male 23:13 * No liver cirrhosis	Apixaban, rivaroxaban	10.8 months (mean) *	Recurrent VTE: *n* = 2 (5.6%) * Major bleeding: *n* = 2 (5.6%) * Clinically relevant non major bleeding: *n* = 1 (2.8%) *
		23 (but only 17 SVT)	Age (mean): 59.8 years * Female/male 6:17 * No liver cirrhosis	LMWH	10.8 months (mean) *	Recurrent VTE: *n* = 3 (13.0%) * Major bleeding: *n* = 3 (13.0%) * Clinically relevant non major bleeding: *n* = 2 (8.7%) *
**Nagaoki (2018)** **[99]**	Retrospective	20	Age (median): 69 years Female/male 7:13 All liver cirrhosis	Edoxaban (30–60 mg OD), preceded by danaparoid	6 months	Recanalization (partial or complete): *n* = 18 (90%) Significant gastrointestinal bleed: *n* = 3 (15%)
		30	Age (median): 67 years Female/male 13:17 All liver cirrhosis	Warfarin (INR target range 1.5–2.0), preceded by danaparoid	6 months	Recanalization (partial or complete): *n* = 9 (30%) Significant gastrointestinal bleed: *n* = 2 (6.7%)
**Hanafy (2019)** **[104]**	Randomized controlled trial	40	Age (mean): 46 years Female/male 8:32 All liver cirrhosis	Rivaroxaban (10 mg BID), preceded by LMWH	Not reported	Recanalization (partial or complete): *n* = 40 (100%) Recurrent SVT: *n* = 0 (0%) Major bleeding: *n* = 0 (0%)
		40	Age (mean): 41.3 years Female/male 5:35 All liver cirrhosis	Warfarin (INR target range 2.0–2.5), preceded by LMWH	Not reported	Recanalization (partial or complete): *n* = 18 (45%) Recurrent SVT: *n* = 4 (10%) Severe upper gastrointestinal bleed: *n* = 17 (42.5%)
**Sharma (2019)** **[100]**	Retrospective	36	Age (median): 29.5 years Female/male 17:19 All BCS patients following endovascular intervention	Dabigatran (150 mg BID), preceded by heparin and/or VKA	10.5 months (mean)	Restenosis: *n* = 4 (11.1%) Major bleeding: *n* = 1 (2.8%) Any bleeding: *n* = 6 (16.7%)
		62	Age (median): 28 years Female/male 24:38 All BCS patients following endovascular intervention	Acenocoumarol (INR target range 2–2.5), preceded by heparin	14.1 months (mean)	Restenosis: *n* = 4 (6.5%) Major bleeding: *n* = 3 (4.8%) Any bleeding: *n* = 7 (11.3%)
**Salim (2019)** **[101]**	Retrospective	22	Age (median): 57.5 years Female/male 7:15 All MVT patients	Apixaban, dabigatran, rivaroxaban	Not reported	Recanalization (partial or complete): *n/N* = 11/16 (68.8%) Major bleeding: *n* = 2 (9.1%) Recurrent SVT: *n* = 0 (0%)
		56	Age (median): 56 years Female/male 21:35 All MVT patients	VKA (INR target range not reported)	Not reported	Recanalization (partial or complete): *n/N* = 29/41 (70.7%) Major bleeding: *n* = 8 (14.3%) Recurrent SVT: *n* = 0 (0%)
		22	Age (median): 65 years Female/male 11:11 All MVT patients	LMWH	Not reported	Recanalization (partial or complete): *n/N* = 6/12 (50%) Major bleeding: *n* = 4 (18.2%) Recurrent SVT: *n* = 0 (0%)
**Naymagon (2020)** **[102]**	Retrospective	93	Age (mean): 47.1 years Female/male 46:47 No liver cirrhosis	Apixaban, dabigatran, rivaroxaban	> 3 months	Recanalization (complete): *n/N* = 61/93 (65.6%) Major bleeding: *n/N* = 2/93 (2.2%)
		108	Age (mean): 50.4 years Female/male 51:57 No liver cirrhosis	Warfarin (INR target range 2.0–3.0)	> 3 months	Recanalization (complete): *n/N* = 33/108 (30.6%) Major bleeding: *n/N* = 26/108 (24.1%)
		70	Age (mean): 51.4 years Female/male 43:27 No liver cirrhosis	LMWH (enoxaparin)	> 3 months	Recanalization (complete): *n/N* = 40/70 (57.1%) Major bleeding: *n/N* = 10/70 (14.3%)

Legend: BCS = Budd–Chiari syndrome; BID = twice daily; DOAC = direct oral anticoagulant; INR = international normalized ratio; LMWH = low molecular weight heparin; MVT = mesenteric vein thrombosis; OD = once daily; SVT = splanchnic vein thrombosis; VKA = vitamin K antagonist. * data not separately reported for SVT patients treated with DOAC or other anticoagulants.

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
