# Peer review of "Cerebral and Splanchnic Vein Thrombosis: Advances, Challenges, and Unanswered Questions"

_jcm, 2020, doi:10.3390/jcm9030743_

Round 1

Reviewer 1 Report

This is a very comprehensive, well-written and thoroughly researched guideline. Almost all aspects of the topic have been covered. Below are some minor comments, in particular, a number of suggestions have been made to improve on spelling and grammar.

Comments

1) It is very common for neurologists to repeat a CT venogram after a few months of treatment for CVT and base duration of anticoagulation on the presence of residual thrombosis. The authors should comment on whether there is any evidence to support this practice.

2) The authors should define Budd Chiari syndrome

3) Spelling/grammar recommendations

‘for’ rather than ‘of’ line 50

‘is’ rather than ‘was’ line 52

‘with’ rather than ‘to’ line 53

‘amongst’ rather than ‘among the’ line 64

‘for’ rather than ‘of’ line 70

‘2.61’ rather than ‘2,61’ line 74

‘has been’ rather than ‘was’ line 78

recommend delete ‘Only’ line 96

‘papillodema’ not ‘papilledema’ line 98

‘the’ can be deleted, line 101

‘attendance at’ rather than ‘an access to’ and ‘prior to’ rather than ‘before’ line 110

Add ‘for ddimer’ at end of line 113

‘patient’ rather than ‘patients’ line 127

‘long time to complete the scan’ line 128

‘is’ rather than ‘are’ line 131

‘has resulted in a’ rather than ‘led towards’ line 134

add ‘a’ between ‘reported’ and ‘mortality’ line 136 and 137

Remove ‘The’ line 139

‘achieved’ rather than ‘obtained’ line 143

‘starting’ rather than ‘to start’ line 154

‘has been’ rather than ‘was’ line 157

‘in’ rather than ‘by’ line 159

‘which’ before ‘reported’ line 165

‘for’ rather than ‘of’ line 170

‘derived’ rather than ‘mediated’ line 211

‘for’ rather than ‘of’ line 240

‘inflammation’ rather than ‘inflammations’ line 244

‘hormonal stimuli also’ rather than ‘also hormonal stimuli’ line 260

‘instead’ is probably redundant in line 286

‘extent’ rather than ‘extension’line 289

‘the’ is probably redundant in line 290

‘difficulty visualising the’ rather than ‘difficult visualization of the’ line 291

‘is’ rather than ‘was’ line 303

‘for’ rather than ‘by’ line 305

The addition of ‘with the risk of further thrombotic events [being the highest…]’ line 313

‘even’ rather than ‘also’ line 317

‘starting’ rather than ‘to start’ line 329

‘non-responsive’ rather than ‘non-respondent’ line 333

‘rates’ rather than ‘rate’ line 363

‘suggest considering’ rather than ‘suggested to consider’ line 380

‘ and cirrhotic patients eligible for liver transplantation’ rather than ‘cirrhotic patients candidate to liver transplant’ line 383

‘suggest considering’ rather than ‘suggested to consider’ line 385

‘recommend’ rather than ‘recommended’ line 387

‘suggesting’ rather than ‘suggested’ line 389

‘has been’ rather than ‘was’ line 403

‘factors’ rather than ‘factor’ line 407

The addition of ‘has’ prior to ‘reduced’ line 408

‘would’ is redundant in line 412

The addition of ‘a’ prior to ‘paucity’ line 418

‘the DOACs’ rather than ‘the DOAC’ line 420

Author Response

Reviewer n.1

This is a very comprehensive, well-written and thoroughly researched guideline. Almost all aspects of the topic have been covered. Below are some minor comments, in particular, a number of suggestions have been made to improve on spelling and grammar.

  • Authors reply: We would like to thank the reviewer for the positive feedback.

Comments

1) It is very common for neurologists to repeat a CT venogram after a few months of treatment for CVT and base duration of anticoagulation on the presence of residual thrombosis. The authors should comment on whether there is any evidence to support this practice.

  • Authors reply: We thank the reviewer for raising this interesting point. A recently published systematic review and meta-analysis (Aguiar de Sousa et al, 2018) reported that recanalization was associated with a favorable neurological outcome; however, evidence of an association between recanalization and CVT recurrence was scarce, thus whether the degree of should impact on anticoagulant treatment duration is still unclear. We clarified this concept in the following sentences (section 3.5 Prognosis): “More than 80% of CVT patients achieved recanalization, either complete or partial, which was associated with a good functional recovery [27,28]. However, it is still unclear whether the lack of recanalization should influence anticoagulant treatment duration, since the evidence on its correlation with CVT recurrence was scarce [28]”.

We also clarified the prognosis of CVT patients by adding the following sentence: “A prognostic score (the CVT risk score) has been proposed to identify patients with a poor prognosis [29]. Six variables were included (malignancy, coma, deep cerebral vein thrombosis – 2 points each; altered mental status, male sex, intracranial hemorrhage – 1 point each) and a score ≥ 3 points showed sensitivity of 96.1% for death or dependency [29].”

2) The authors should define Budd Chiari syndrome

  • Authors reply: We thank the reviewer for this suggestion. The Budd-Chiari syndrome has now been defined (section 5.1 Definition and epidemiology) “SVT refers to thrombosis occurring in the veins draining the abdominal viscera, and includes portal vein thrombosis (PVT), mesenteric veins thrombosis (MVT), splenic vein thrombosis and the Budd-Chiari syndrome (BCS) [47]. BCS refers to the obstruction of the hepatic venous outflow, which can be located at any site between the small hepatic venules and the confluence of the inferior vena cava into the right atrium [48].”

3) Spelling/grammar recommendations

‘for’ rather than ‘of’ line 50

‘is’ rather than ‘was’ line 52

‘with’ rather than ‘to’ line 53

‘amongst’ rather than ‘among the’ line 64

‘for’ rather than ‘of’ line 70

‘2.61’ rather than ‘2,61’ line 74

‘has been’ rather than ‘was’ line 78

recommend delete ‘Only’ line 96

‘papillodema’ not ‘papilledema’ line 98

‘the’ can be deleted, line 101

‘attendance at’ rather than ‘an access to’ and ‘prior to’ rather than ‘before’ line 110

Add ‘for ddimer’ at end of line 113

‘patient’ rather than ‘patients’ line 127

‘long time to complete the scan’ line 128

‘is’ rather than ‘are’ line 131

‘has resulted in a’ rather than ‘led towards’ line 134

add ‘a’ between ‘reported’ and ‘mortality’ line 136 and 137

Remove ‘The’ line 139

‘achieved’ rather than ‘obtained’ line 143

‘starting’ rather than ‘to start’ line 154

‘has been’ rather than ‘was’ line 157

‘in’ rather than ‘by’ line 159

‘which’ before ‘reported’ line 165

‘for’ rather than ‘of’ line 170

‘derived’ rather than ‘mediated’ line 211

‘for’ rather than ‘of’ line 240

‘inflammation’ rather than ‘inflammations’ line 244

‘hormonal stimuli also’ rather than ‘also hormonal stimuli’ line 260

‘instead’ is probably redundant in line 286

‘extent’ rather than ‘extension’line 289

‘the’ is probably redundant in line 290

‘difficulty visualising the’ rather than ‘difficult visualization of the’ line 291

‘is’ rather than ‘was’ line 303

‘for’ rather than ‘by’ line 305

The addition of ‘with the risk of further thrombotic events [being the highest…]’ line 313

‘even’ rather than ‘also’ line 317

‘starting’ rather than ‘to start’ line 329

‘non-responsive’ rather than ‘non-respondent’ line 333

‘rates’ rather than ‘rate’ line 363

‘suggest considering’ rather than ‘suggested to consider’ line 380

‘ and cirrhotic patients eligible for liver transplantation’ rather than ‘cirrhotic patients candidate to liver transplant’ line 383

‘suggest considering’ rather than ‘suggested to consider’ line 385

‘recommend’ rather than ‘recommended’ line 387

‘suggesting’ rather than ‘suggested’ line 389

‘has been’ rather than ‘was’ line 403

‘factors’ rather than ‘factor’ line 407

The addition of ‘has’ prior to ‘reduced’ line 408

‘would’ is redundant in line 412

The addition of ‘a’ prior to ‘paucity’ line 418

‘the DOACs’ rather than ‘the DOAC’ line 420

  • Authors reply: We apologize for the grammar and spelling mistakes. Corrections have been made as suggested.

Reviewer 2 Report

This is a well-written and clearly organized narrative review on CVT and SVT by two experts in the field. I have several minor comments for improvement.

  1. Please consider including splanchnic vein thrombosis and cerebral vein thrombosis in the title to clearly label the focus of the article. The term unusual site thrombosis includes many other vascular beds, which are not addressed in this review.

  1. Clinical case 1 – Please specify that the patient was taking a combination OCP (as opposed to a progesterone-only pill).

  1. Page 2, line 69 – Similarly, please specify what you mean by hormonal treatments. I assume you mean exogenous estrogens.

  1. Page 2, line 81, “however, thrombophilia testing might help to tailor anticoagulant treatment duration, and therefore prevent future VTE” – I have some concerns about this conjecture. It presupposes that the finding of an underlying thrombophilia may affect risk of recurrent VTE. The evidence for this after DVT or PE is equivocal. Is there evidence after CVT (or SVT for that matter)? I think we need to be careful about suggesting courses of action based on thrombophilia test results without good evidence.

  1. The authors provide a good summary of published evidence and guidelines on treatment of CVT. Since they are international experts in this area and evidence is limited, it would be welcome if they added a brief description of how they treat CVT. Do they typically start with LMWH? Do they use DOACs or only warfarin? How do they determine duration of anticoagulation?

  1. Section 5.2 – Another underlying condition that predisposes to SVT, especially BCS, is paroxysmal nocturnal hemoglobinuria (PNH). Screening for PNH is common in patients who present with SVT. Please include information on PNH in this section.

  1. Similar to comment 5, it would be very welcome if the authors briefly described their approach to management of SVT. Do they use DOACs? How do they determine duration of anticoagulation?

Author Response

Reviewer n.2

This is a well-written and clearly organized narrative review on CVT and SVT by two experts in the field. I have several minor comments for improvement.

  • Authors reply: We would like to thank the reviewer for the positive feedback.

  1. Please consider including splanchnic vein thrombosis and cerebral vein thrombosis in the title to clearly label the focus of the article. The term unusual site thrombosis includes many other vascular beds, which are not addressed in this review.
  • Authors reply: We thank the reviewer for this suggestion. The title has been changed as follows “Cerebral and splanchnic vein thrombosis: advances, challenges and unanswered questions”

  1. Clinical case 1 – Please specify that the patient was taking a combination OCP (as opposed to a progesterone-only pill).
  • Authors reply: We thank the reviewer for this comment. Clinical case 1 has been updated as follows (section 2) “She had recently started a combined estrogen-progestin oral contraceptive pill.”

  1. Page 2, line 69 – Similarly, please specify what you mean by hormonal treatments. I assume you mean exogenous estrogens.
  • Authors reply: We thank the reviewer for this suggestion. We updated section 3.2 as follows “Systemic risk factors are commonly found in CVT. Sex-specific risk factors, such as pregnancy/puerperium and hormonal treatments (such as oral contraceptives or estrogen replacement treatment), are reported in 10-17% and 50-53% of women with CVT, respectively [2, 3].”

  1. Page 2, line 81, “however, thrombophilia testing might help to tailor anticoagulant treatment duration, and therefore prevent future VTE” – I have some concerns about this conjecture. It presupposes that the finding of an underlying thrombophilia may affect risk of recurrent VTE. The evidence for this after DVT or PE is equivocal. Is there evidence after CVT (or SVT for that matter)? I think we need to be careful about suggesting courses of action based on thrombophilia test results without good evidence.
  • Authors reply: We recognize that the above-mentioned statement might have been misleading. Evidence on the association between thrombophilia and recurrent CVT is scarce, however the management of severe thrombophilic disorders can be derived from usual site VTE. We have clarified as follows (section 3.2. Risk factors): “It has been suggested not to routinely screen all CVT patients for occult malignancy, due to the low prevalence of this condition, or thrombophilia, given the limited clinical relevance. However, in accurately selected patients (such as those with young age, unprovoked CVT or family history of CVT) [14] finding a severe thrombophilia (i.e. antithrombin, protein C or S deficiencies, or homozygosity for factor V Leiden or prothrombin G20210A mutation) would suggest an indefinite anticoagulant treatment duration, similarly to usual site VTE.”

We also clarified with regards to SVT (Section 5.2 Risk factors) “After excluding major local and systemic risk factors, thrombophilia testing is suggested in patients with high pretest probability of severe thrombophilia (such as those with young age, unprovoked SVT, or family history of VTE).”

  1. The authors provide a good summary of published evidence and guidelines on treatment of CVT. Since they are international experts in this area and evidence is limited, it would be welcome if they added a brief description of how they treat CVT. Do they typically start with LMWH? Do they use DOACs or only warfarin? How do they determine duration of anticoagulation?
  • Authors reply: We thank the reviewer for this suggestion. We have added the following paragraph in Section 6 (Expert Opinion) “In general, we start anticoagulant treatment for CVT in the acute phase with LMWH (or UFH in patients with severe renal failure). UFH may also be preferred in patients with intracranial hemorrhage and severe clinical presentation. After clinical stability, the majority of patients are switched to VKA. Pending more evidence on the use of the DOACs and based on the favorable results of the RE-SPECT CVT trial [44], we consider the use of dabigatran in selected patients with good general health conditions and low bleeding risk. However, the DOACs are currently off label for the treatment of unusual VTE. A minimum anticoagulant treatment duration of 3 months is usually prescribed, to be extended up to 6-12 months for unprovoked CVT and indefinite for recurrent VTE or persistent risk factors (such as severe thrombophilia).”

  1. Section 5.2 – Another underlying condition that predisposes to SVT, especially BCS, is paroxysmal nocturnal hemoglobinuria (PNH). Screening for PNH is common in patients who present with SVT. Please include information on PNH in this section.
  • Authors reply: We thank the reviewer for this comment. Section 5.2 has been updated as follows “MPN represent the most common systemic risk factor, being present in 40.9% of BCS and 31.5% of non-malignant non-cirrhotic PVT [64, 65]. In particular, the JAK2 V617F mutation was reported in 37-45% of BCS and 24-34% of PVT [66, 67]. Paroxysmal nocturnal hemoglobinuria (PNH) is another hematological disorder which can predispose to the development of SVT, particularly BCS [68]. However, the prevalence of PNH among SVT patients is <1% [69].”

  1. Similar to comment 5, it would be very welcome if the authors briefly described their approach to management of SVT. Do they use DOACs? How do they determine duration of anticoagulation?
  • Authors reply: We thank the reviewer for this suggestion. We have added the following paragraph in Section 6 (Expert Opinion) “After assessing the risk of bleeding, we usually start the anticoagulant treatment for SVT in the acute phase with LMWH, together with adequate prophylaxis of variceal bleeding, if needed. In patients with thrombocytopenia, a reduced dose of LMWH might be required. We consider UFH in patients with recent bleeding, severe renal failure, or potential candidate for surgery (such as those with extensive bowel infarct). After clinical stability, we switch the majority of patients to VKA, with the exception of those with cancer associated SVT, severe thrombocytopenia or severe liver disease. Pending more evidence on the use of the DOACs and based on the results of RCTs conducted on usual site VTE [105-108], we consider the use of factor-Xa inhibitors in selected patients without severe liver or renal impairment and at low bleeding risk. The majority of SVT patients are candidate for long-term anticoagulation, having either a major persistent risk factor, an unprovoked SVT or a BCS; however, we favor a shorter duration of 3-6 months in those with transient risk factors (such as recent abdominal surgery or abdominal inflammation/infections).